# Short- and Mid-Term Improvement of Postural Balance after a Neurorehabilitation Program via Hippotherapy in Patients with Sensorimotor Impairment after Cerebral Palsy: A Preliminary Kinetic Approach

**DOI:** 10.3390/brainsci9100261

**Published:** 2019-09-29

**Authors:** Hélène Viruega, Inès Gaillard, John Carr, Bill Greenwood, Manuel Gaviria

**Affiliations:** 1Institut Equiphoria, Combo Besso—Rouges Parets, 48500 La Canourgue, France; helene.viruega@equiphoria.com (H.V.); ines.gaillard@equiphoria.com (I.G.); 2Racewood Ltd., 15A Park Road, Tarporley, Cheshire CW6 0AN, UK; john@racewood.com (J.C.); bill.greenwood@racewood.com (B.G.)

**Keywords:** postural balance, cerebral palsy, neural plasticity, neurorehabilitation, hippotherapy, horse riding simulator

## Abstract

There is still a lack of studies focused on trunk neurorehabilitation. Accordingly, it is unclear which therapeutic modalities are the most effective in improving static/dynamic balance after brain damage. We designed a pilot study on hippotherapy to assess its short- and mid-term effect on dynamic postural balance in patients with moderate-to-severe sensorimotor impairment secondary to cerebral palsy. Five patients aged 15.4 ± 6.1 years old were recruited. All of them had moderate-to-severe alterations of the muscle tone with associated postural balance impairment. Standing and walking were also impaired. Ten minutes horse riding simulator followed by twenty minutes hippotherapy session were conducted during five session days separated by one week each. We analyzed the displacement of the Center of Pressure (COP) on the sitting surface of the simulator’s saddle by means of a customized pressure pad. We measured the general behavior of the COP displacement as well as the postural adjustments when pace changed from walk to trot to walk during the sessions and among sessions. Statistical analysis revealed an improved postural control both by the end of the session and from session 1 to session 5. These results suggest that hippotherapy might support regularization of postural control in a long-term neurorehabilitation context.

## 1. Introduction

Postural control is the intrinsic ability to restore balance from any position or during any motor activity resulting in a final motor adjustment process [1]. Integrating data coming from visual, vestibular, proprioceptive and tactile inputs, drives posture-regulating muscles in the whole body, especially in the trunk [2]. Postural reactions are usually considered automatic; actually, they are somewhere between reflexive and volitional actions. They involve activation of muscle synergies throughout the body, depend on prior experience and may change according to task and context [3]. Postural control is therefore a complex task driven by the central nervous system involving anticipatory predictive reactions for balance’s maintenance through the harmonious interaction of muscle activity against gravity and environmental forces [4,5].

In neurological disorders such as cerebral palsy, the presence of muscle tone changes, muscle weakness, musculoskeletal alterations, decreased shoulder and pelvic girdle movement, makes sitting postural control precarious and unsafe. The integration of posture awareness of the upper trunk and shoulder complex as a stable basis for upper limb movement is an essential component for rehabilitation [6,7,8,9]. 

The use of unstable support surfaces in trunk rehabilitation results in increased muscular activity during the execution of trunk exercises improving muscle strength, endurance, trunk flexibility, dynamic balance and proprioception [10,11,12]. A change of surface during trunk rehabilitation might be a good strategy to enhance static and dynamic balance. However, the benefits might not outweigh the possible disadvantages, namely increased muscle fatigue and fear of falling [13].

Hippotherapy is an emerging specialized rehabilitation approach, performed on a horse under the direction of accredited health professionals (e.g., medical doctors, physical therapists, occupational therapists, psychomotricians, speech-language pathologists, psychologists). The horse’s movement at the walk is used as a therapeutic support by the therapist [14]. Main published evidence of hippotherapy efficacy on postural balance improvement has been described in the field of neurorehabilitation of conditions such as multiple sclerosis [15], autism [16], stroke [17], cerebral palsy [18,19,20,21] or movement disorders [22]. 

The horse simulator has been used as useful complement for hippotherapy in posture and gait rehabilitation [21,23,24,25,26,27,28]. It allows to overcome the variability of a hippotherapy session for scientific purposes, i.e., horse start and stop, irregular pace depending on environment and stimuli, intersession variability. It brings safe, regular and sustained stimuli to patients while facilitating a more homogeneous evaluation. 

Essentially, methods used to characterize postural control changes rely on clinical examination before and after treatment. They include the Berg Balance Scale [17,29,30,31], Pediatric Balance Scale [20,21], Performance Oriented Mobility Assessment [17,30], Gross Motor Function Measure [18,23,32], Sitting Assessment Scale [32], Bruininks-Oseretsky Test of Motor Proficiency [16], Timed Up and Go test [29,33]. Also, several quantitative measures have been implemented such as video-based movement analysis on a static barrel after hippotherapy [19], or on a riding simulator [28], force platforms in static standing position before and after hippotherapy or riding simulation [21,31,33], pressure sensitive mat in sitting position before and after hippotherapy or riding simulation [23].

Only one study in healthy volunteers has quantitatively measured the dynamic postural behavior during riding on a simulator [28]. No studies aiming to characterize the displacement of the Center of Pressure (COP) in sitting position during horse-riding simulation have been yet published. The present pilot study aims at the preliminary evaluation of the efficacy of an experimental protocol of horse-riding simulator and hippotherapy in short-term dynamic postural balance evolution in patients with sensorimotor impairment secondary to cerebral palsy.

## 2. Materials and Methods

### 2.1. Subjects and Inclusion Criteria

Eligible patients included males and non-pregnant females, from 5 to 25 years old, presenting a sensorimotor impairment secondary to cerebral palsy, able to understand the basis of the experimental protocol. Patients were included based on their current rehabilitation management in our Institute. The experimental protocol was part of their therapeutic program.

The inclusion criteria were as follows:Diagnosis: CP consisting of spastic tetraparesis, diparesis, monoparesis or hemiparesisDegree of impairment according to GMFCS: levels II to IV (Walking with Limitations to Self-Mobility with Limitations/May Use Powered Mobility)Good comprehension ability (no cognitive or behavioral impairment)Normal hip joint function and sufficient abduction to allow ridingNo concomitant pathology that may impair sensorimotor and/or cognitive functionNo history of treatment with botulinum toxin in the 6 months preceding the start of the protocolNo therapeutic intervention planned during the duration of the study (injection of botulinum toxin, tenotomy, tendon transfer, etc.)No history of uncontrolled painCertificate of non-contraindication issued by the responsible physician

Patients were not included if they disagreed to participate to the study. Patients were free to withdraw at any time without any consequence on the quality of their care. All subjects and/or their home institutions gave written informed consent in accordance with the Declaration of Helsinki.

The pilot study was carried out in agreement with the recommendations of the regional ethics committee of Montpellier-France (CPP Sud-Mediterranée IV; n°150403 dated 06/09/2015) that provided ethical approval, and institutional and national guidelines for research in human individuals (ANSM approval n°151108B31 dated 09/02/2015). All subjects and/or their home institutions gave written informed consent in accordance with the Declaration of Helsinki.

### 2.2. Study Design

We used a single-group repeated-measures longitudinal prospective pilot study to evaluate the effects of an experimental protocol of horse-riding simulator and hippotherapy in postural balance efficacy in patients with sensorimotor impairment secondary to cerebral palsy. For this purpose, a single experiment consisting of 10 minutes simulator prior to the 20 minutes hippotherapy session was carried out during five session days. We measure the postural strategy, from a kinetic point of view, in day one and five and during the first and last two minutes of the riding simulation. For this end, we recorded the sagittal (anterior-posterior) and coronal (lateral) displacement of the COP by means of sensors embedded in the saddle of the simulator. The experimental procedure consisted of modifying the simulator’s pace and recording the signals as follows:(1)during the first two minutes: 60 s walk—20 s trot—20 s walk—20 s trot(2)during the last two minutes: 60 s walk—30 s trot—30 s walk

Sample size calculation (GLIMMPSE 2.0.0 open access software) was based on 90% power for a comparison of one-group repeated measures with a type I error rate of 0.05 using the Hotelling-Lawley Trace multivariate approach for a CV of 10% and 50% in the coronal and sagittal plane respectively based on Clayton [34]. The total sample size was *n* = 5. 

Two main outcomes were evaluated:Short term effect comparing postural balance at the beginning and at the end of the 10-minute simulator’s exercise (*n* = 5)Mid term effect comparing the remaining effect after 5 sessions, separated by one week each, on (i) postural balance evolution (at the beginning of the session) and (ii) efficacy (adjustment speed during the session) (*n* = 5)

Regular intervention of cerebral palsy patients at the institute, usually coming from medical-social centers where they live permanently, consists of one-hour weekly session for 12–16 weeks. They take care of their horse, participate in their equipment and spend up to 30 minutes on the walking horse surrounded by a team of three people (a horse manager, a physiotherapist or a psychologist and a trained accompanying person or another practitioner). Evidence-based hippotherapy practice has not been standardized and is quite heterogeneous, i.e., from 6 to 12 weeks in weekly, bi-weekly or tri-weekly bases [18,19,22,23].

We introduced the simulator as an intermediate step to prepare the patient to the hippotherapy sessions. The goal was to enhance postural balance and to dissociate the pelvic girdle from the lumbar column and lower limbs. Also, the simulator helped warm-up the patients for a better physiological performance [35], and soften apprehension to prevent negative stress effects on cognitive functioning [36]. Ten minutes on the simulator seems to be clinically effective on the majority of our patients without the onset of fatigue. Again, the evidence-based practice on a simulator is significantly heterogeneous, i.e., from 10 to 40 min [17,23,30,31,37]. In the present study, the simulator was set to walk in-between the two recording intervals (first and the last two minutes) to prevent fatigue.

### 2.3. Outcome Measures

Therapeutic Equine Simulator System (TESS, Racewood Ltd., Tarporley, UK) is a horse simulator, custom made for Equiphoria (Figure 1a,b), which includes the biomechanical characteristics of five of Equiphoria’s horses dedicated to physical therapy. Three different paces have been incorporated, namely walk, trot and canter. TESS aims to provide a vast array of information, including force applied to the reins, sitting pressure and legs contact pressure.

For the purpose of the study, only the sitting pressure was activated and recorded. The saddle sensors consist of four air pressure sensors, embedded in a flat rectangular matrix of 440 mm width × 520 mm length connected to electronic circuit boards. This is then connected to the central unit by means of shielding cables. Conductivity varies according to the pressure applied. Saddle sensors are specially customized, the saddle directly sits upon leather and foam pads containing four airbags. The sensors measure the pressure transmitted by the air displacement in the bags. There is an electronic control unit which converts the pressure in the airbags to an electrical voltage and passes this information to a central computer. The information gathered from the four sensors is then correlated to show weight distribution as a COP. This is graphically represented on an orthogonal plane on the simulator’s screen for biofeedback purposes.

The COP values are expressed in mm. Saddle X measures the pressure in the left/right direction (coronal plane) and Y in the forward/backward direction (sagittal plane). Negative X value corresponds to a bias to left side and positive X value towards right. Negative Y value corresponds to a bias towards the rear and positive Y value towards the front. Time is expressed in seconds (see Figure 1c).

Feasibility studies were conducted for a year prior to the current experimental protocol since we used a custom method to assess the displacement of COP.

The COP is defined as the center point of force in the *x* direction and *y* direction that a participant exerts on the pressure pad when sitting; this movement is displayed as a traveling dot between the trunk that moves with weight shift. Two COP variables were selected considering that the starting point of COP is different among patients and because of their clinical relevance, mathematical integrity, and suitability for clinical evaluation: the path length-per-second (*P*) and the average radial displacement (*R_d_*) [38]. 

Path length-per-second (*P*) is the average distance traveled per second during the time period of one sample, where *f* is the sampling rate per second, *N* is the number of sample points, *x_i_* is the instantaneous center of pressure in the mediolateral direction, and *y_i_* is the instantaneous center of pressure in the anterior–posterior direction, at sample index *i*:(1)$$P=\frac{f\times \sum_{i-1}^{N-1}\sqrt{\left({\left(x_{i}+1-x_{i}\right)}^{2}+{\left(y_{i}+1-y_{i}\right)}^{2}\right)}}{N-1}$$

Average radial distance (*R_d_*) is the average of the instantaneous COP radial distance:(2)$$R_{d}=\frac{\sum_{i=1}^{N}\sqrt{\left({\left(x_{i}-x_{c}\right)}^{2}+{\left(y_{i}-y_{c}\right)}^{2}\right)}}{N}$$

Every sample point (*x_i_*, *y_i_*) has an associated radial displacement from the centroid (*x_c_*, *y_c_*). The mediolateral position center (*x_c_*) and the anterior-posterior position center (*y_c_*) are defined as:(3)$$x_{c}=\frac{\sum_{i=1}^{N}\left(x_{i}\right)}{N},\;\mathrm{and}\;y_{c}=\frac{\sum_{i=1}^{N}\left(y_{i}\right)}{N}$$

### 2.4. Measurement Accuracy

The software for signal treatment was customized by Racewood to provide X and Y raw data (in Volts) of the displacement of the COP. The values range from 0 V to 10 V with a fluctuation of ±0.05–0.1 V to the voltage signal inputs when running the system due to the nature of the electronics/cables and proximity to high powered electronic motors. The sampling frequency is 30 Hz, enough for capturing COP displacement during the most dynamic movement such as trot and canter. The maximal recording time is 60 min. The beginning and the end of the recording are triggered manually. Data are written in CSV format and stored in the central unit for delayed treatment.

### 2.5. Data Analysis

The displacement of COP in the two anatomical planes was analyzed to demonstrate the short term effect (intrasession changes) by:Comparing the postural balance behavior at different times within each session: first two minutes of session 1 versus last two minutes of session 1 and first two minutes of session 5 versus last two minutes of session 5;Comparing the evolution of the compensatory postural adjustments (CPA) or feedback mechanisms (activated by sensory events following loss of stable posture) during 3 seconds after each change of pace (T0, T60, T80, T100, T480, T540 and T570) within each session;Comparing the evolution of the anticipatory postural adjustments (APA) or feedforward mechanisms (predicting disturbances and producing preprogrammed responses that maintain stability) during 3 seconds before each change of pace (T57, T77, T97, T117, T537, T567 and T597) within each session;Comparing the compensatory versus the anticipatory postural adjustment (CPA versus APA) around each pace change, i.e., 3 seconds before and after the change (T57 vs. T60, T77 vs. T80, T97 vs. T100, T537 vs. T540, T567 vs. T570) within each session.

The displacement of COP was also analyzed to demonstrate the mid term effect (intersession changes) by:Comparing the general shape of the displacement of session 1 versus session 5 (the first and last two minutes) to have a general picture of the overall mid-term effect of the therapy on dynamic postural balance;Comparing the postural balance behavior at corresponding times: first two minutes of session 1 versus last two minutes of session 1 versus first two minutes of session 5 and versus last two minutes of session 5;Comparing the evolution of the compensatory postural adjustments (CPA) during 3 seconds after each change of pace (T0, T60, T80, T100, T480, T540 and T570) at corresponding times of session 1 and 5;Comparing the evolution of the anticipatory postural adjustments (APA) during 3 seconds before each change of pace (T57, T77, T97, T117, T537, T567 and T597) at corresponding times of session 1 and 5.

### 2.6. Statistical Analysis

Values are presented as means and standard deviations (SD). Two types of tests were used for comparing data. A Wilcoxon matched-pairs signed rank test allowed comparing the general shape of displacement of COP among sessions 1 and 5. A Friedmann test followed by a Dunn’s multiple comparison post hoc test was applied to determine the differences of the overall balance behavior with respect to time (displacement of the COP at minute 1 and 8 of the first and last session) and the evolution of the postural balance with respect to changes of the external input (pace changes) and to time (compensatory postural adjustments and anticipatory postural adjustments versus time). A P level of 0.05 was considered statistically significant. GraphPad Prism 8 software (GraphPad Software Inc., San Diego, CA, USA) was used for statistical analysis.

## 3. Results

### 3.1. Study Population

Five subjects (two females, three males, age 15.4 ± 6.1 years) diagnosed with cerebral palsy were included in this study. The procedure was part of the patient’s therapeutic 12–16 weeks program at our institute. The subjects were recruited according to the inclusion criteria and to their clinical picture (Table 1). All of them had moderate-to-severe alterations of the muscle tone with associated impairment of posture and balance, hindering their ADL. Standing and walking were also impaired at various degrees. During the study period, the patients that participate did not follow any other therapeutic intervention.

### 3.2. General Shape of the Displacement of COP between Session 1 and 5

The overall COP path length mean value at session 1 was 201.7 (±65.2) mm/s. At session 5, it was 193.9 (±72.9) mm/s. The mean COP path length at session 1 was significantly different against session 5 (*p* < 0.001). 

The overall COP ARD mean value at session 1 was 91.9 (±16.6) mm. At session 5, it was 37.2 (±12.0) mm. The mean COP ARD at session 1 was significantly different against session 5 (*p* < 0.001) (Figure 2).

### 3.3. COP Behavior at the Different Times

At session 1, the COP path length during the first 2 minutes ranged 68.2 mm/s to 702.9 mm/s with a mean value of 216.4 (±69.4) mm/s. During the last 2 minutes, COP path length ranged 51.8 mm/s to 1078.0 mm/s with a mean value of 187.0 (±57.1) mm/s. At session 5, the COP path length during the first 2 minutes ranged 49.1 mm/s to 641.8 mm/s with a mean value of 203.6 (±82.4) mm/s. During the last 2 minutes, COP path length ranged 58.3 mm/s to 519.9 mm/s with a mean value of 184.3 (±60.4) mm/s.

Concerning the COP ARD, it ranged 62.9 mm to 154.0 mm during the first 2 minutes of session 1 with a mean value of 102.2 (±16.3) mm. It ranged 58.7 mm to 116.6 mm during the last 2 minutes with a mean value of 81.7 (±8.6) mm. At session 5, the COP ARD during the first 2 minutes ranged 16.0 mm to 104.3 mm with a mean value of 42.3 (±12.1) mm. During the last 2 minutes, COP ARD ranged 11.4 mm to 76.4 mm with a mean value of 32.1 (±9.6) mm.

Short term effect:

At session 1, the mean path length of the COP during the first 2 minutes was significantly different against the last 2 minutes (216.4 mm/s vs. 187.0 mm/s; *p* < 0.001). At session 5, the mean path length of the COP during the first 2 minutes was also significantly different against the last 2 minutes (216.4 mm/s vs. 187.0 mm/s; *p* < 0.001) (see Figure 3a).

Also at session 1, the mean ARD of the COP during the first 2 minutes was significantly different against the last 2 minutes (102.2 mm vs. 81.7 mm; *p* < 0.001). At session 5, the mean ARD of the COP during the first 2 minutes was significantly different against the last 2 minutes (42.3 mm vs. 32.1 mm; *p* < 0.001) (see Figure 3b).

Mid term effect:

When comparing the last 2 minutes of session 1 against the first 2 minutes of session 5, the COP path length was significantly different but higher in session 5 (187.0 mm/s versus 203.6 mm; *p* < 0.001) (see Figure 3a). The COP ARD decresed from session 1 to session 5 (81.7 mm vs. 42.3 mm) and was also significantly different (*p* < 0.001) (see Figure 3b).

Lastly, when comparing the first 2 minutes of session 1 against the last 2 minutes of session 5, the COP path length and the COP ARD were significantly different (216.4 mm/s vs. 184.3 mm/s and *p* < 0.001; 102.2 mm vs. 32.2 mm and *p* < 0.001 respectively) (Figure 3a,b).

### 3.4. Behavior of the Compensatory Postural Adjustments (CPA) with Pace Changes

#### 3.4.1. Between Successive Pace Changes within the Same Sequence

CPA was often challenged during the ten minutes-sequence at sessions 1 and 5 when pace changed: at the beginning from stop to walk at T0 (Walk 1), from walk to trot at T60 (Trot 1), from trot to walk at T80 (Walk 2), from walk to trot at T100 (Trot 2), at the beginning of the second recording after a brief stop to walk at T480 (Walk 3), from walk to trot at T540 (Trot 3), and from trot to walk at T570 (Walk 4). Consecutive pace changes (3 seconds CPA intervals, i.e., 90 frames) were compared two-by-two for COP path length and ARD (Friedmann test followed by Dunn’s multiple comparisons test).

Throughout session 1, COP path length during CPA was not significantly different when comparing the pace changes from the beginning to the end (Table 2 and Figure 4a). Also, COP ARD during CPA was not significantly different when comparing the pace changes from the beginning to the end of the session except for T80 vs. T100 (Table 2 and Figure 4b). During session 5, COP path length during CPA was significantly different (*p* > 0.001) when comparing the pace changes from T0 vs. T60, T80 vs. T100 and T540 vs. T570 (Table 2 and Figure 4a). COP ARD during CPA was significantly different (*p* > 0.001) when comparing the pace changes from T0 vs. T60, T100 vs. T480 and T540 vs. T570 (see Table 2 and Figure 4b). 

#### 3.4.2. Between Corresponding Times of Session 1 and 5 when Pace Changes

Corresponding pace changes (3 seconds CPA intervals after change) were compared two-by-two (Friedmann test followed by Dunn’s multiple comparisons test) among sessions 1 and 5. COP path length during CPA was not significantly different among sessions (Table 3 and Figure 5a). Conversely, COP ARD during CPA was significantly different among sessions (*p* < 0.001; Table 3 and Figure 5b).

### 3.5. Behavior of the Anticipatory Postural Adjustments (APA) before Pace Changes

#### 3.5.1. Between Successive Pace Changes within the Same Sequence

APA was often challenged during the ten minutes-sequence at session 1 and 5 before pace change: from walk to trot at T57, from trot to walk at T77, from walk to trot at T97, before the end of the first two minutes recording time (brief stop) at T117, from walk to trot at T537, from trot to walk at T567 and finally before stop at T597. Consecutive pace changes (3 s APA intervals before change, i.e., 90 frames) were compared two-by-two for COP path length and ARD (Friedmann test followed by Dunn’s multiple comparisons test).

Throughout session 1, COP path length during APA was significantly different (*p* < 0.001) when comparing the pace changes from the beginning to the end (Table 4 and Figure 4c). Conversely, COP ARD during APA was not significantly different when comparing the pace changes from the beginning to the end of the session except for T77 vs T97 and T97 vs T117 (Table 4 and Figure 4d). During session 5, COP path length during APA was significantly different (*p* > 0.001) from the beginning to the end (Table 4 and Figure 4c). COP ARD during APA was significantly different (*p* > 0.001) when comparing the pace changes from T537 vs. T567 and T567 vs. T597 (see Table 4 and Figure 4d). 

#### 3.5.2. Between Corresponding Times of Session 1 and 5 when Pace Changes

Corresponding pace changes (3 seconds APA intervals before pace change) were compared two-by-two (Friedmann test followed by Dunn’s multiple comparisons test) among sessions 1 and 5. COP path length during APA was not significantly different among sessions (Table 5 and Figure 5c) except for T567 (*p* < 0.05). Conversely, COP ARD during APA was significantly different among sessions whatever the time of pace change (*p* < 0.001; Table 5 and Figure 5d). 

### 3.6. Compensatory Postural Adjustment (CPA) versus Anticipatory Postural Adjustment (APA)

Comparison of COP path length and COP average radial displacement during CPA versus APA did not allow to establishing significant differences among the two different postural adjustment mechanisms (Table 6 and Figure 6). As a whole, the dispersion of the data was substantially equivalent.

## 4. Discussion

The present study aimed to investigate the contribution of the horse-riding simulator combined with hippotherapy, i.e., an unstable support surface approach for trunk neurorehabilitation, to the overall improvement of dynamic postural control in patients with sensorimotor impairment secondary to cerebral palsy. In such patients, defective generation of motor patterns, such as delayed onset of muscle activation and contraction, unsuitable amplitude or abnormal activation sequence, hinders the sophisticated mechanisms of posture regulation [6,7,8]. To maintain balance and stability, postural responses to perturbations must be appropriately scaled to how fast and how far the body mass center is displaced on the support base [3,4].

Postural rehabilitation has been a rich domain of innovation throughout the last decades. Mainly relying on physical therapy [39], it has progressively been enriched by new knowledge on brain functioning and technical/technological aids. Thus, biofeedback in the 1980s and 1990s [40] was enriched one decade later by reactive balance training using platform perturbations and anticipatory balance training with computer feedback [6,8]. Subsequently, hippotherapy, treadmill training, upper limb therapy, and strength training were introduced [7,14,18,41,42], followed in the early 2000s by trunk-targeted training, reactive balance training and gross motor task training [8]. In the current decade, some other methods have been developed and complete the therapeutic arsenal for postural rehabilitation such as functional electrical stimulation [43], virtual reality training [44], whole body vibration training [45] and robot-assisted training [46].

Hippotherapy is a multisensory activity in which the rhythmic and three-dimensional sway stimulates the patient’s postural reflex mechanisms, resulting in balance and coordination reinforcement [33]. Hippotherapy requires whole body involvement and thus contributes to the development of strength, muscle tone, flexibility, relaxation, body awareness and enhanced motor coordination and balance helping to gain a more normative sense of body symmetry [18,40]. The rhythmically oscillating back of the horse mainly stimulates the rider’s postural reflex mechanisms, resulting in solicitation of balance and coordination. 

The horse’s locomotion and the simulator’s motion trigger a forward to backwards’ movement, generating anterior and posterior pelvic tilt. This stimulates trunk stability via flexor and extensor muscles. In addition, the lateral movement results in the reciprocal activation of trunk’s lateral flexors and can help to further reinforce its stability. The rotating component of the movement induces trunk rotation, most likely resulting in lateral flexors’ activation [41]. Altogether, the constant overall movement of the simulator generates a series of volitional-free micro-adjustments of the patient’s body.

In our study, the trunk stability via the flexor and extensor muscles and the lateral flexors was explored indirectly through the displacement of the projection of the COP on the sitting surface. Indeed, no EMG recording was done to confirm our hypothesis and the trunk stability in this children could be ensured by other compensating mechanisms. However, in general terms, the absolute COP showed a shift towards the centre of the sitting support (Cartesian coordinate system), reflected by a significant decrease of the COP path length and COP average radial displacement noticed both during the first and fifth session. Interestingly, the comparison of the COP at the end of the first session with the beginning of the fifth session showed a significant decrease of the average radial displacement value consistent with a consolidation of postural improvement. 

Automatic postural responses depend on the influence of central drive on the generated motor response. Descending commands from this central drive prepare sensory and motor systems in order to anticipate the response and to speed up and optimize the motor adjustment to the stimulus. The disadvantage is that the central command can produce errors in the motor responses when the stimulus or the external condition changes unexpectedly [47] or in case of damaged central drive [3]. Thus, postural fine-tuning is achieved by means of two major mechanisms: (i) compensatory or feedback mechanisms activated by sensory events following loss of desirable posture (compensatory postural adjustments), and (ii) anticipatory or feed-forward mechanisms predicting disturbances and producing preprogrammed responses that maintain stability (anticipatory postural adjustments). When analyzing the anticipatory postural adjustments in our cohort of patients, we observed a clear trend towards stabilization of posture close to the midline through a muscle strategy most likely involving the trunk’s flexor and extensor muscles. This was revealed by the values of COP average radial displacement decreasing decreasing by around 50%, with a remarkable steadiness all along session five around 55 mm. Even though the trends were similar for the compensatory postural adjustments, the COP average radial displacement showed more variability. The onset of automated postural responses arises prior to voluntary control and the characteristics of those responses are different from that of voluntary movements [48]. However, higher brain levels within the cerebral cortex can modulate postural responses by changing the activity of the pathways involved in their generation [49]. Motor cortex provides a critical contribution to postural control [50]. Data from studies in humans demonstrated that inhibition of the motor cortex can reduce postural activity of trunk muscles [51]. As cortical regions contribute to postural control, it is plausible that deficits in postural activation, such as those observed in people with cerebral palsy, may be associated with changes in the excitability and motor cortex organization and vice versa [52]. 

Motor and sensory cortices have a huge potential to operate changes in their organization as described in brain conditions [47,53]. Reorganization of the motor cortex often engages multiple steps along different time scales. In a short time scale, expansion of motor maps into a deafferented cortex is done by unmasking the latent intracortical connections between the two regions [54]. Subsequently in the long term, activity-dependent plastic changes such as growth of new horizontal connection [55] or synaptogenesis [56] have been highlighted. The cortex has been identified by others as the primary substrate for remodelling [57].

The global organizational changes would be to a certain extent driven by injury and rehabilitation intervention that could be crucial for functional recovery. After brain damage, although there are spontaneous reparative mechanisms that follow the injury, these are hardly ever sufficient to support substantial functional recovery [58]. The degree of plastic changes is related to both the relevance of an experience and the intensity or frequency of its constituent events [58]. According to previous works, experiences that are highly relevant are likely to produce much more rapid neuronal changes than less relevant ones. In contrast, experiences that are perceived as irrelevant may not lead to neural changes. In the same way, intensity or frequency of the experience is also crucial [59,60]. Since self-awareness is not deeply altered in the patients participating to this pilot, one can reasonably assume that strong psychological cues are playing a key role in making relevant the experience on the horse and the simulator [61]. Current findings suggest that improved postural skills in general can be associated with the plasticity of the CNS, and that spinal and supraspinal adjustments are responsible for postural control’s enhancement. The results observed in this study on postural improvement after only 5 sessions would indicate a possible short to mid time scale for such changes. 

## 5. Conclusions

Notwithstanding the crucial role of balance on movement and motor function, there is still a lack of studies that focus on postural control and rehabilitation. Accordingly, it is unclear which modalities of training are the most effective in improving static and dynamic balance. Stable and unstable support surfaces have been used accounting for positive results [62], but current rehabilitation programs tend to be tedious, resource-intensive and require dedicated facilities [63]. Therefore, there is an urgent need to identify and develop innovative, reliable, safe and user-friendly intervention methods.

It has been suggested that unstable support surfaces induce increased activation of the trunk musculature and a constant muscle response in order to adjust the posture against instability [62]. Marshall suggested that muscle activity increased when the centre of mass is further away from the unstable support surface [64]. Therefore, patients do not need to carry out complex exercise routines to benefit from trunk rehabilitation on unstable support surfaces. Our horse riding simulator generates here up to hundred three-dimensional physical smooth trunk and girdle movements by minute mimicking that of the patient’s body during hippotherapy. This has a prominent role in promoting functional recovery of postural balance in neurological diseases [15,20,22].

The present study suggests that hippotherapy combined with riding simulator intervention might support regularization of postural control in the short-to-mid term rehabilitation context. Nevertheless, a more solid scientific groundwork is needed in particular the correlation with clinical postural tests. This would improve our understanding on movement integration in brain injured patients and potentially facilitate the patients’ rehabilitation process. The current method represents an encouraging opportunity for neurorehabilitation of postural balance impairment.

## 6. Study Limitations and Perspectives

The current pilot was designed to evaluate the effects of an experimental protocol of horse-riding simulation combined with hippotherapy on postural balance efficacy in patients with sensorimotor impairment secondary to cerebral palsy. The cohort was small and rather heterogeneous and a larger cohort study has to confirm and fine-tune our encouraging preliminary results. Also, it could be relevant to couple the kinetic measures with kinematics and/or electromyographic data in order to have a more complete picture of the relationship between the central drive and the effectors. Furthermore, a long term follow up on the residual changes after the termination of the hippotherapy cycle has to be performed in future studies.

## Figures and Tables

**Figure 1 brainsci-09-00261-f001:**
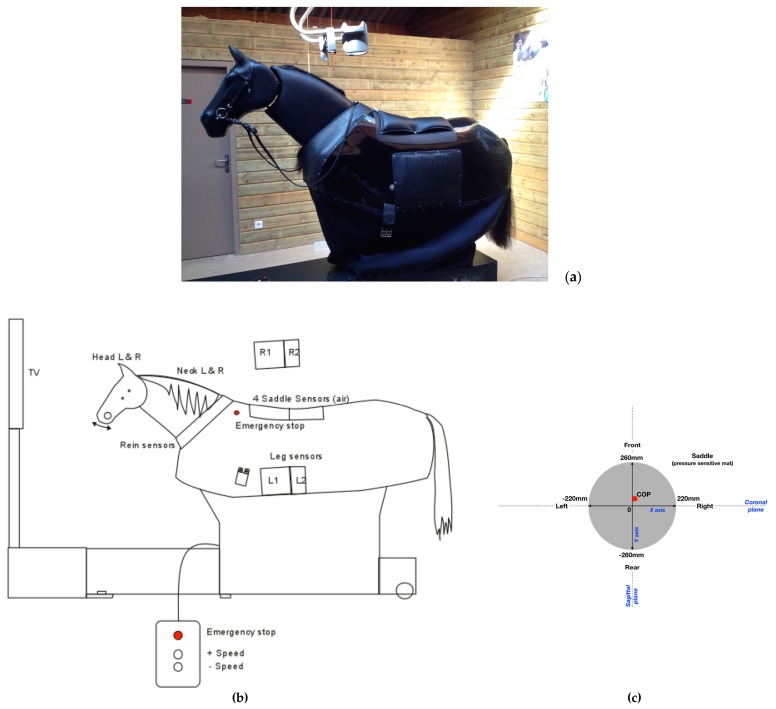
(**a**) Photograph of the equine simulator; (**b**) Diagram of the equine simulator used in the hippotherapeutic treatment; (**c**) Diagram of the pressure sensitive saddle and the analyzed biomechanical variables.

**Figure 2 brainsci-09-00261-f002:**
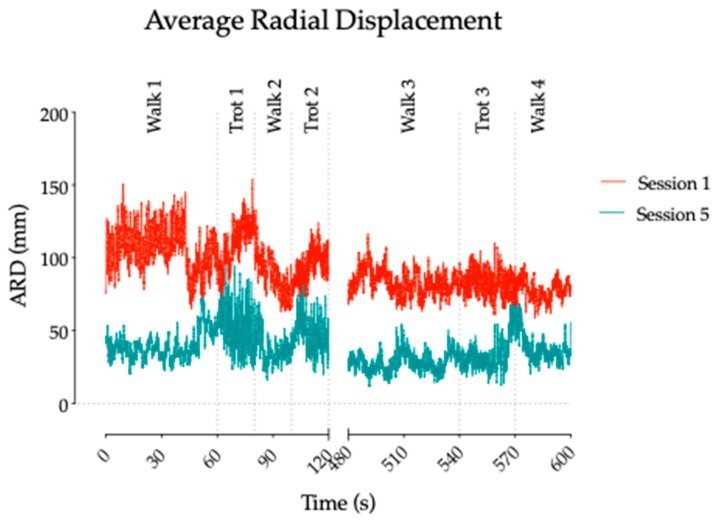
General evolution of the COP Average radial displacement (ARD) for sessions 1 and 5. The figure shows the different pace changes during the acquisition (first 2 minutes and last 2 minutes).

**Figure 3 brainsci-09-00261-f003:**
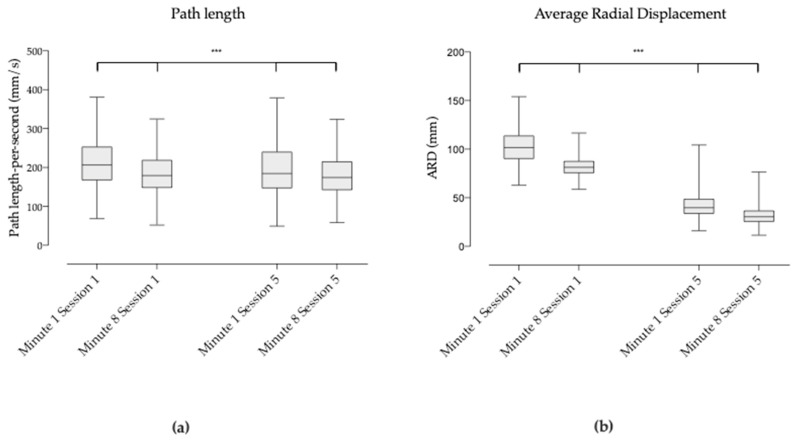
(**a**) Box and whiskers diagram of the COP path length in mm/s of the first two min. versus the last two min. of sessions 1 and 5; (**b**) Box and whiskers diagram of the COP Average radial displacement (ARD) in mm of the first two min. versus the last two min. of sessions 1 and 5.

**Figure 4 brainsci-09-00261-f004:**
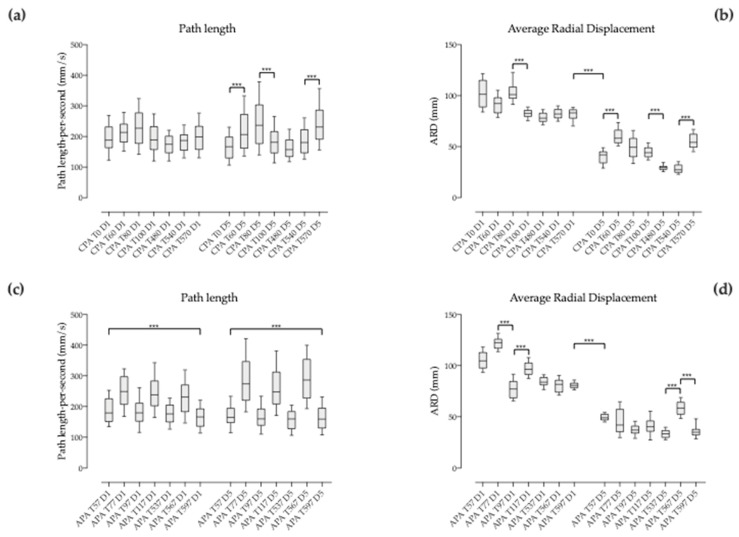
Box and whiskers diagram corresponding to intrasession comparison of: (**a**) COP path length during compensatory postural adjustment (CPA) and (**b**) COP average radial displacement during CPA at different times within each session during the three seconds that follow each pace change; (**c**) COP path length during anticipatory postural adjustment (APA) and (**d**) COP average radial displacement during APA at different times within each session during the three seconds before each pace change. D1 corresponds to session 1 and D5 to session 5.

**Figure 5 brainsci-09-00261-f005:**
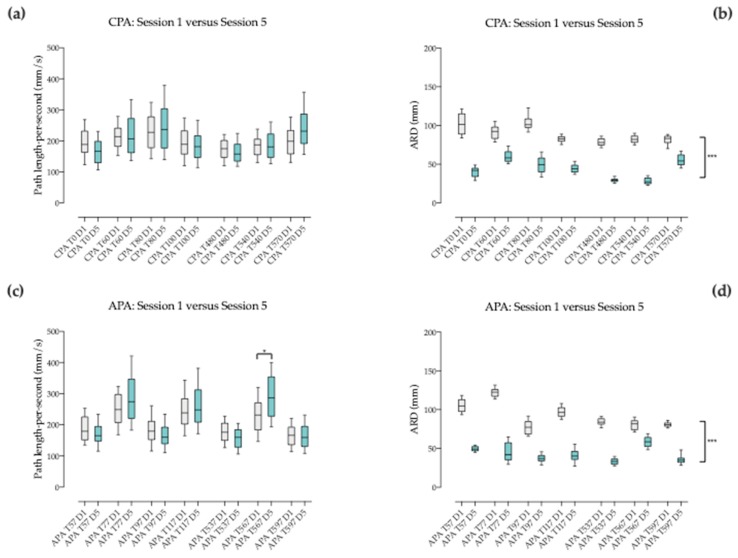
Box and whiskers diagrams corresponding to the comparisons of COP path length (**a**) and COP average radial displacement (**b**) during compensatory postural adjustments (CPA) and the comparisons of COP path length (**c**) and COP average radial displacement (**d**) during anticipatory postural adjustments (APA) between corresponding pace changes of sessions 1 and 5. D1 corresponds to session 1 and D5 to session 5. *p* < 0.05 (*), *p* < 0.001 (***).

**Figure 6 brainsci-09-00261-f006:**
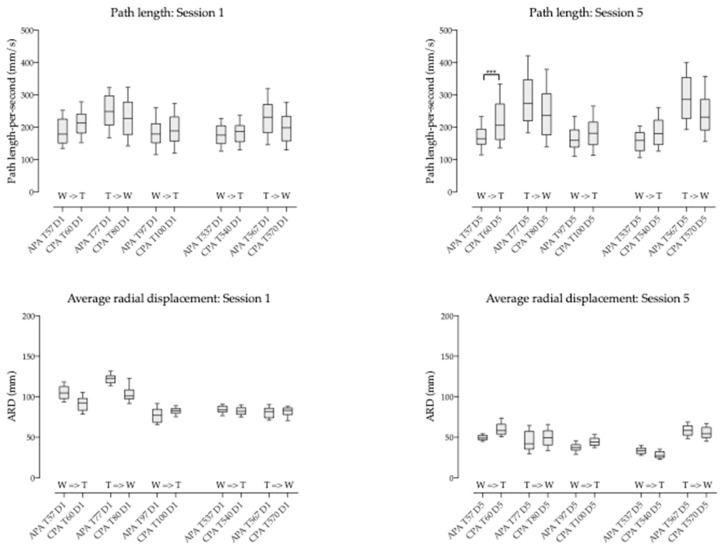
Comparison of consecutive compensatory versus anticipatory postural adjustments (CPA versus APA) for the two measured variables (COP path length and COP ARD) at different times during both sessions.

**Table 1 brainsci-09-00261-t001:** Clinical characteristics of the recruited population.

Age	Sex	Motor deficit	GMFCS	Axial Hypotonia *	Sitting Postural Deficit **	Adductor’s Hypertonia ***
12	M	spastic tetraparesis	IV	+	++	2
24	F	spastic tetraparesis	III	++	+	2
19	M	spastic tetraparesis	IV	++	++	2
8	M	spastic tetraparesis	III	++	++	1+
14	F	tetraparesis	II	+++	++	0

* History and physical exam (decreased muscle strength, decreased activity tolerance, delayed motor skills development, rounded shoulder posture, hypermobile joints/increased flexibility); ** Able to sit less than 60 seconds under supervision (+), less than 30 seconds under supervision (++); *** Modified Ashworth Scale.

**Table 2 brainsci-09-00261-t002:** Comparison of intrasession COP path length and COP average radial displacement during compensatory postural adjustment (CPA) when pace changes (3 second intervals). Values correspond to means ± SD of the mean. *p* < 0.001 (***).

COP Path Length during CPA	Value 1	Value 2	*p* Value
**Session 1**			
T0 (Walk 1) vs. T60 (Trot 1)	196.1 ± 56.1	215.9 ± 56.1	ns
T60 (Trot 1) vs. T80 (Walk 2)	215.9 ± 56.1	229.3 ± 71.2	ns
T80 (Walk 2) vs. T100 (Trot 2)	229.3 ± 71.2	196.9 ± 54.9	ns
T100 (Trot 2) vs. T480 (Walk 3)	196.9 ± 54.9	172.4 ± 39.6	ns
T480 (Walk 3) vs. T540 (Trot 3)	172.4 ± 39.6	183.9 ± 44.2	ns
T540 (Trot 3) vs. T570 (Walk 4)	183.9 ± 44.2	200.6 ± 58.5	ns
**Session 5**			
T0 (Walk 1) vs. T60 (Trot 1)	167.9 ± 48.4	219.3 ± 75.2	***
T60 (Trot 1) vs. T80 (Walk 2)	219.3 ± 75.2	254.5 ± 98.1	ns
T80 (Walk 2) vs. T100 (Trot 2)	254.5 ± 98.1	185.8 ± 58.4	***
T100 (Trot 2) vs. T480 (Walk 3)	185.8 ± 58.4	163.5 ± 42.1	ns
T480 (Walk 3) vs. T540 (Trot 3)	163.5 ± 42.1	187.4 ± 49.2	ns
T540 (Trot 3) vs. T570 (Walk 4)	187.4 ± 49.2	241.4 ± 73.6	***
**COP average radial displacement during CPA**			
**Session 1**			
T0 (Walk 1) vs. T60 (Trot 1)	102.2 ± 14.4	91.3 ± 9.5	ns
T60 (Trot 1) vs. T80 (Walk 2)	91.3 ± 9.5	103.9 ± 11.6	ns
T80 (Walk 2) vs. T100 (Trot 2)	103.9 ± 11.6	82.4 ± 6.0	***
T100 (Trot 2) vs. T480 (Walk 3)	82.4 ± 6.0	78.6 ± 5.4	ns
T480 (Walk 3) vs. T540 (Trot 3)	78.6 ± 5.4	82.4 ± 6.0	ns
T540 (Trot 3) vs. T570 (Walk 4)	82.4 ± 6.0	81.5 ± 6.5	ns
**Session 5**			
T0 (Walk 1) vs. T60 (Trot 1)	40.1 ± 7.2	60.3 ± 8.7	***
T60 (Trot 1) vs. T80 (Walk 2)	60.3 ± 8.7	49.1 ± 11.3	ns
T80 (Walk 2) vs. T100 (Trot 2)	49.1 ± 11.3	44.8 ± 6.1	ns
T100 (Trot 2) vs. T480 (Walk 3)	44.8 ± 6.1	29.5 ± 3.0	***
T480 (Walk 3) vs. T540 (Trot 3)	29.5 ± 3.0	28.2 ± 4.5	ns
T540 (Trot 3) vs. T570 (Walk 4)	28.2 ± 4.5	55.6 ± 8.4	***

**Table 3 brainsci-09-00261-t003:** Comparison of COP path length and COP average radial displacement values during compensatory postural adjustment (CPA) between corresponding times of session 1 and 5 when pace changes (3 seconds interval after change). D1 corresponds to session 1 and D5 to session 5. Values correspond to means ± SD of the mean. *p* < 0.001 (***).

COP Path Length during CPA	Session 1	Session 5	*p* Value
T0 D1 (Walk 1) vs. T0 D5 (Walk 1)	196.1 ± 56.1	167.9 ± 48.4	ns
T60 D1 (Trot 1) vs. T60 D5 (Trot 1)	215.9 ± 56.1	219.3 ± 75.2	ns
T80 D1 (Walk 2) vs. T80 D5 (Walk 2)	229.3 ± 71.2	254.5 ± 98.1	ns
T100 D1 (Trot 2) vs. T100 D5 (Trot 2)	196.9 ± 54.9	185.8 ± 58.4	ns
T480 D1 (Walk 3) vs. T480 D5 (Walk 3)	172.4 ± 39.6	163.5 ± 42.1	ns
T540 D1 (Trot 3) vs. T540 D5 (Trot 3)	183.9 ± 44.2	187.4 ± 49.2	ns
T570 D1 (Walk 4) vs. T570 D5 (Walk 4)	200.6 ± 58.5	241.4 ± 73.6	ns
**COP average radial displacement during CPA**			
T0 D1 (Walk 1) vs. T0 D5 (Walk 1)	102.2 ± 14.4	40.1 ± 7.2	***
T60 D1 (Trot 1) vs. T60 D5 (Trot 1)	91.3 ± 9.5	60.3 ± 8.7	***
T80 D1 (Walk 2) vs. T80 D5 (Walk 2)	103.9 ± 11.6	49.1 ± 11.3	***
T100 D1 (Trot 2) vs. T100 D5 (Trot 2)	82.4 ± 6.0	44.8 ± 6.1	***
T480 D1 (Walk 3) vs. T480 D5 (Walk 3)	78.6 ± 5.4	29.5 ± 3.0	***
T540 D1 (Trot 3) vs. T540 D5 (Trot 3)	82.4 ± 6.0	28.2 ± 4.5	***
T570 D1 (Walk 4) vs. T570 D5 (Walk 4)	81.5 ± 6.5	55.6 ± 8.4	***

**Table 4 brainsci-09-00261-t004:** Comparison of intrasession COP path length and COP average radial displacement during anticipatory postural adjustment (APA) when pace changes (3 second intervals before changing). Values correspond to means ± SD of the mean. *p* < 0.001 (***).

COP Path Length during APA	Value 1	Value 2	*p* Value
**Session 1**			
T57 D1 vs. T77 D1	188.6 ± 46.7	250.9 ± 62.0	***
T77 D1 vs. T97 D1	250.9 ± 62.0	184.5 ± 49.6	***
T97 D1 vs. T117 D1	184.5 ± 49.6	247.2 ± 66.4	***
T117 D1 vs. T537 D1	247.2 ± 66.4	177.7 ± 40.2	***
T537 D1 vs. T567 D1	177.7 ± 40.2	232.3 ± 67.4	***
T567 D1 vs. T597 D1	232.3 ± 67.4	167.3 ± 41.8	***
**Session 5**			
T57 D5 vs. T77 D5	169.6 ± 40.3	290.8 ± 94.2	***
T77 D5 vs. T97 D5	290.8 ± 94.2	166.5 ± 44.0	***
T97 D5 vs. T117 D5	166.5 ± 44.0	268.2 ± 92.1	***
T117 D5 vs. T537 D5	268.2 ± 92.1	157.5 ± 40.3	***
T537 D5 vs. T567 D5	157.5 ± 40.3	289.4 ± 82.1	***
T567 D5 vs. T597 D5	289.4 ± 82.1	164.5 ± 46.5	***
**COP average radial displacement during APA**			
**Session 1**			
T57 D1 vs. T77 D1	104.9 ± 9.5	122.8 ± 8.6	ns
T77 D1 vs. T97 D1	122.8 ± 8.6	77.3 ± 9.2	***
T97 D1 vs. T117 D1	77.3 ± 9.2	96.8 ± 7.4	***
T117 D1 vs. T537 D1	96.8 ± 7.4	84.1 ± 5.1	ns
T537 D1 vs. T567 D1	84.1 ± 5.1	80.5 ± 7.2	ns
T567 D1 vs. T597 D1	80.5 ± 7.2	80.7 ± 3.6	ns
**Session 5**			
T57 D5 vs. T77 D5	49.5 ± 3.8	45.5 ± 13.6	ns
T77 D5 vs. T97 D5	45.5 ± 13.6	37.3 ± 5.8	ns
T97 D5 vs. T117 D5	37.3 ± 5.8	41.0 ± 10.4	ns
T117 D5 vs. T537 D5	41.0 ± 10.4	33.5 ± 4.6	ns
T537 D5 vs. T567 D5	33.5 ± 4.6	58.3 ± 7.8	***
T567 D5 vs. T597 D5	58.3 ± 7.8	36.1 ± 6.5	***

**Table 5 brainsci-09-00261-t005:** Comparison of COP path length and COP average radial displacement values during anticipatory postural adjustment (APA) between corresponding times of session 1 and 5 when pace changes (3 seconds interval before change). D1 corresponds to session 1 and D5 to session 5. Values correspond to means ± SD of the mean. *p* < 0.05 (*), *p* < 0.001 (***).

COP Path Length during APA	Session 1	Session 5	*p* Value
T57 D1 vs. T57 D5	188.6 ± 46.7	169.6 ± 40.3	ns
T77 D1 vs. T77 D5	250.9 ± 62.0	290.8 ± 94.2	ns
T97 D1 vs. T97 D5	184.5 ± 49.6	166.5 ± 44.0	ns
T117 D1 vs. T117 D5	247.2 ± 66.4	268.2 ± 92.1	ns
T537 D1 vs. T537 D5	177.7 ± 40.2	157.5 ± 40.3	ns
T567 D1 vs. T567 D5	232.3 ± 67.4	289.4 ± 82.1	*
T597 D1 vs. T597 D5	167.3 ± 41.8	164.5 ± 46.5	ns
**COP average radial displacement during APA**			
T57 D1 vs. T57 D5	104.9 ± 9.5	49.5 ± 3.8	***
T77 D1 vs. T77 D5	122.8 ± 8.6	45.5 ± 13.6	***
T97 D1 vs. T97 D5	77.3 ± 9.2	37.3 ± 5.8	***
T117 D1 vs. T117 D5	96.8 ± 7.4	41.0 ± 10.4	***
T537 D1 vs. T537 D5	84.1 ± 5.1	33.5 ± 4.6	***
T567 D1 vs. T567 D5	80.5 ± 7.2	58.3 ± 7.8	***
T597 D1 vs. T597 D5	80.7 ± 3.6	36.1 ± 6.5	***

**Table 6 brainsci-09-00261-t006:** Comparison of compensatory postural adjustment (CPA) versus anticipatory postural adjustment (APA) for the two measured variables (COP path length and COP ARD) during pace changes. D1 corresponds to session 1 and D5 to session 5. Values correspond to means ± SD of the mean. *p* < 0.001 (***).

COP Path Length	APA	CPA	*p* value
**Session 1**			
T57 (walk) vs. T60 (trot)	188.6 ± 46.7	215.9 ± 56.1	ns
T77 (trot) vs. T80 (walk)	250.9 ± 62.0	229.3 ± 71.2	ns
T97 (walk) vs. T100 (trot)	184.5 ± 49.6	196.9 ± 54.9	ns
T537 (walk) vs. T540 (trot)	177.7 ± 40.2	183.9 ± 44.2	ns
T567 (trot) vs. T570 (walk)	232.3 ± 67.4	200.6 ± 58.5	ns
**Session 5**			
T57 (walk) vs. T60 (trot)	169.6 ± 40.3	219.3 ± 75.2	***
T77 (trot) vs. T80 (walk)	290.8 ± 94.2	254.5 ± 98.1	ns
T97 (walk) vs. T100 (trot)	166.5 ± 44.0	185.8 ± 58.4	ns
T537 (walk) vs. T540 (trot)	157.5 ± 40.3	187.4 ± 49.2	ns
T567 (trot) vs. T570 (walk)	289.4 ± 82.1	241.4 ± 73.6	ns
**COP average radial displacement**			
**Session 1**			
T57 (walk) vs. T60 (trot)	104.9 ± 9.5	91.3 ± 9.5	ns
T77 (trot) vs. T80 (walk)	122.8 ± 8.6	103.9 ± 11.6	ns
T97 (walk) vs. T100 (trot)	77.3 ± 9.2	82.4 ± 6.0	ns
T537 (walk) vs. T540 (trot)	84.1 ± 5.1	82.4 ± 6.0	ns
T567 (trot) vs. T570 (walk)	80.5 ± 7.2	81.5 ± 6.5	ns
**Session 5**			
T57 (walk) vs. T60 (trot)	49.5 ± 3.8	60.3 ± 8.7	ns
T77 (trot) vs. T80 (walk)	45.5 ± 13.6	49.1 ± 11.3	ns
T97 (walk) vs. T100 (trot)	37.3 ± 5.8	44.8 ± 6.1	ns
T537 (walk) vs. T540 (trot)	33.5 ± 4.6	28.2 ± 4.5	ns
T567 (trot) vs. T570 (walk)	58.3 ± 7.8	55.6 ± 8.4	ns

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
