# Peer review of "Short- and Mid-Term Improvement of Postural Balance after a Neurorehabilitation Program via Hippotherapy in Patients with Sensorimotor Impairment after Cerebral Palsy: A Preliminary Kinetic Approach"

_brainsci, 2019, doi:10.3390/brainsci9100261_

Round 1

Reviewer 1 Report

Short- and mid-term improvement of postural balance after a neurorehabilitation program via hippotherapy

General comments:

I like the idea of the equine simulator and I think it has the potential to provide valuable data.

The data analysis is flawed because it assumes that all participants start each session in the exact same position, which is impossible. Rather than using x and y positions of the center of pressure, the authors need to look at some measures of the dispersion of the COP. Path length and average radial distance would be valid measures to evaluate.

The text in all figures is too small to read and font size needs to be increased. I couldn’t read the text in Figures 1-4 at all.

Specific comments:

Methods

It would be helpful in Section 2.5 to provide categories (short term effect and mid-term effect) for the analyses described. It’s rather unclear as is. Please explain how CPA and APA were calculated. Please explain that CPA and APA were compared at Session 1, Session 5, and between Sessions 1 and 5. Do you have any photographs of the equine simulator? It would be a nice addition.

Results

Please explain in Methods the classification/scale for axial hypotonia and sitting postural deficit in Table 1. There is no reason to include standing and walking ability descriptions in Table 1 because they are redundant and less precise than GMFCS. Sections 3.2 and 3.3 are flawed because COP on the Cartesian plane is meaningless. It is dependent on starting position. The center of pressure is simply defined as the center point of force in the x and y directions that a subject exerts on a force plate (or in this case, the pressure sensitive saddle) during a given activity; this movement is displayed as a traveling point that moves with weight shift. What you are interested in is the dispersion of COP. Specifically, path length and average radial distance (ARD). Path length is the average distance the center of pressure travels over each trial. ARD is the mean radial distance of the center of pressure from the centroid over the entire trial. These center of pressure reference formulae have greater clinical relevance, mathematic integrity, and are suitable for clinical evaluation using a force platform.

References:

Wolff D, Rose J, Jones V, Bloch D, Oehlert J, Gamble J. Postural balance measurements for normal children and adolescents. J Orthop Res 1998;16:271-5.

Rose J, Wolff D, Jones V, Bloch D, Oehlert J, Gamble J. Postural balance in children with cerebral palsy. Dev Med Child Neurol 2002;44:58-63.

The Friedman test can be used to determine whether path length and ARD values changed within participants over time.

Tables 2-5 are challenging to read/interpret and have errors in the headings. For example, the headings for Tables 2 and 4 would be more meaningful if Mean value 1 and Mean value 2 stated Beginning of Session and End of Session, respectively. Table 4 has the variable FPA; do you mean APA? Why not say Walk 1 vs. Trot 1, Trot 1 vs. Walk 2, etc.? D1 versus D5 CPA and APA are meaningless comparisons because you can’t know whether the individual started in the exact same position both times. Again, path length and ARD would alleviate this problem.

Discussion

Please add a reference to lines 359-360. Without the use of electromyography, it is not valid to state that your participants were using their trunk flexor/extensor muscles (lines 385-386). They may have been compensating in some unknown way. There should be a section on limitations, namely that there was a very small number of participants and that they were heterogeneous in their distribution and type of cerebral palsy.

Author Response

General comments:

Data analysis has been improved by using COP path length-per-second and COP average radial displacement as suggested by the reviewer.

The text of figures has been adapted in order to easily read it.

Methods
In Section 2.5 the two categories, short term and mid term effect, were provided with respect to described analyses.

CPA and APA time framework was clarified.

A picture of the simulator was provided (Figure 1a).

Results

Table 1 was modified and classification of sitting postural deficit and hypotonia were clarified. The column containing the data for stand and walk was deleted since the information is redundant with GMFCS.

The COP is now measured through the Path length-per-second (P) and the Average radial displacement (ARD) which are independent variables with respect to the absolute position of the patient on the pressure pad.

The reference has been added (Rose J, Wolff D, Jones V, Bloch D, Oehlert J, Gamble J. Postural balance in children with cerebral palsy. Dev Med Child Neurol 2002;44:58-63).

The ANOVA test has been replaced by the non parametric Friedmann test and the t-test by the non parametric Wilcoxon test taking into account the small sample size.

Table headings have been simplified as much as it could be done and errors were corrected considering the comments of the reviewer.

CPA and APA were analyzed through COP path length and COP ARD to take into account the different position of each patient at the beginning of the trial.

Discussion
The assumption of the use of trunk flexor/extensor muscles has been put into perspective given the absence of evidence by EMG.

A section Study limitations and perspectives has been included after the Conclusion.

Reviewer 2 Report

Thank you for giving me the opportunity to review this manuscript.

The authors tried to show the effect of hippotherapy using their own unique method. It could be analyzed the displacement of the center of the pressure in sitting position during horse-riding simulation. I acknowledge that the authors have performed a relatively new study, with new data measurements and analyses.

However, I could not understand how the COP position was calculated and what the measurement unit of the value was. I think if the authors knew the positions of saddle sensors, a COP could be calculated by the pressure force of the sensors. In that case, “mm” or “cm” in SI unit should be employed to express the displacement. It looks strange to express the COP position using a percentage.

There is another point of concern. The authors used a mean value of the COP during two minutes of sessions for the sagittal direction. If participants were sitting on the saddle in exactly the same position for all sessions, it is OK to think anterior shift of the mean COP indicates that the participant sat leaning forward. However, if the participant could move his buttock on the saddle, the anterior displacement of the COP might simply indicate that the participant was sitting in front of the saddle. The authors should explain more detail about this issue.

Author Response

The COP has been better defined in « 2.3 Outcome measures ».

The COP position is now expressed in mm taking into account the total area of the pressure pad (orthogonal plane of 440mm width x 520mm length).

Two variables have been defined considering that the starting point of COP is different among patients: (i) the COP Path length-per-second (P) and (ii) the COP Average radial displacement (Rd) (Rose et al., 2002); these parameters are considered relevant and accurate for clinical and biomechanical evaluation of postural balance through force platforms and are independent from the absolute position of the patient on the platform.

Rose J, Wolff D, Jones V, Bloch D, Oehlert J, Gamble J. Postural balance in children with cerebral palsy. Dev Med Child Neurol 2002;44:58-63.

Round 2

Reviewer 1 Report

All of my concerns have been addressed. Thank you for a nice paper